# Prophylactic ICD Survival Benefit Prediction: Review and Comparison between Main Scores

**DOI:** 10.3390/jcm13175307

**Published:** 2024-09-07

**Authors:** Moshe Rav-Acha, Ziv Dadon, Arik Wolak, Tal Hasin, Ilan Goldenberg, Michael Glikson

**Affiliations:** 1Jesselson Integrated Heart Center Share Zedek Medical Center, Jerusalem 9103102, Israel; ziv.dadon@mail.huji.ac.il (Z.D.); arikwt@szmc.org.il (A.W.); hasint@szmc.org.il (T.H.); mglikson@szmc.org.il (M.G.); 2Faculty of Medicine, Hebrew University, Jerusalem 9112102, Israel; 3Department of Medicine, University of Rochester Medical Center, New York, NY 14627, USA; ilan.goldenberg@heart.rochester.edu

**Keywords:** primary prevention, ICD, survival, prediction

## Abstract

Current guidelines advocate for the use of prophylactic implantable cardioverter defibrillators (ICDs) for all patients with symptomatic heart failure (HF) with low ejection fraction (EF). As many patients will never use their device and some are prone to device-related complications, scoring systems for delineating subgroups with differential ICD survival benefits are crucial to maximize ICD benefit and mitigate complications. This review summarizes the main scores, including MADIT trial-based Risk Stratification Score (MRSS) and Seattle Heart Failure Model (SHFM), which are based on randomized trials with a control group (HF medication only) and validated on large cohorts of ‘real-world’ HF patients. Recent studies using cardiac MRI (CMR) to predict ventricular arrhythmia (VA) are mentioned as well. The review shows that most scores could not delineate sustained VA incidence, but rather mortality without prior appropriate ICD therapies. Multiple scores could identify high-risk subgroups with extremely high probability of early mortality after ICD implant. On the other hand, low-risk subgroups were defined, in whom a high ratio of appropriate ICD therapy versus death without prior appropriate ICD therapy was found, suggesting significant ICD survival benefit. Moreover, MRSS and SHFM proved actual ICD survival benefit in low- and medium-risk subgroups when compared with control patients, and no benefit in high-risk subgroups, consisting of 16–20% of all ICD candidates. CMR reliably identified areas of myocardial scar and ‘channels’, significantly associated with VA. We conclude that as for today, multiple scoring models could delineate patient subgroups that would benefit differently from prophylactic ICD. Due to their modest-moderate predictability, these scores are still not ready to be implemented into clinical guidelines, but could aid decision regarding prophylactic ICD in borderline cases, as elderly patients and those with multiple co-morbidities. CMR is a promising technique which might help delineate patients with a low- versus high-risk for future VA, beyond EF alone. Lastly, genetic analysis could identify specific mutations in a non-negligible percent of patients, and a few of these mutations were found to predict an increased arrhythmic risk.

## 1. Introduction

Based on pivot studies performed two decades ago [1,2], current guidelines recommend primary prevention implantable cardioverter defibrillator (ICD) implantation for all patients with symptomatic (NYHA class II-III) heart failure (HF) with both ischemic and non-ischemic cardiomyopathies (ICM and NICM, respectively) with left ventricular ejection fraction (LVEF) ≤ 35% after 3 months of optimal medical therapy (OMT) [3,4]. Recent ESC guidelines extend the recommendation for NICM patients with LVEF 36–50% and two other risk factors for sudden cardiac death (SCD), including unexplained syncope, inducible sustained monmorphic VT on electrophysiological study (EPS), the presence of late gadolinium enhancement on CMR, and pathogenic mutations associated with increased VA such as Lamin A/C and Filamin C mutations [3]. The ACC/AHA guidelines for NICM are very similar except for a IIb indication for prophylactic ICD in NYHA I patients with LVEF ≤ 35%, which is not indicated according to ESC guidelines. As for ICM, both ESC and ACC/AHA guidelines recommend a prophylactic ICD also for NYHA I patients with LVEF ≤ 30% as well as for post-MI patients with LVEF ≤ 40% who had a spontaneous non-sustained VT and inducible monomorphic VT on EPS (both have a class I recommendation in ACC/AHA and IIa in ESC guidelines) [3,4]. Importantly, prophylactic ICD should NOT be implanted in patients with an early (<1-year) expected mortality, as given in the ESC guidelines and emphasized as class III indication in the ACC/AHA guidelines. Nevertheless, the efficacy of this strategy is questionable as most sudden cardiac death victims have normal LVEF [5] and many primary prevention candidates will never use their ICD in their lifetime [6,7,8]. Indeed, in routine clinical practice only 25–37% of primary prevention ICD patients experience potentially life-saving ICD intervention in the first five years after implantation [9,10]. Moreover, the number of prophylactic ICDs needed to save one life was reported to be 24, as many of these HF patients die due to non-arrhythmic causes including end-stage HF itself and other co-morbidities which are common in these patients [11]. Thus, these guidelines were criticized as many patients may suffer from ICD-related complications (lead revisions, device endocarditis, etc.) without any survival benefit, in addition to the financial burden on limited health care systems imposed by these guidelines [6,8]. Accordingly, various scores were developed to delineate HF subgroups that would benefit significantly or minimally from prophylactic ICD. The current narrative review will summarize the main relevant scores with an emphasis on their strengths and weaknesses; the strength of their database, highlighting scores which were based on pivot randomized trials comparing survival between HF patients treated by OMT only versus OMT+ICD to enable a direct actual proof of ICD survival benefit; and the validation of these scores in real world large cohorts. A suggested use of the various scores to aid clinical decision making regarding prophylactic ICD implant is shown (Graphical Abstract). Notably, no systemic bibliographic search was used for this review. Lastly, we will also refer to genetic mutations found to imply an increased ventricular arrhythmia and SCD risk and to the recent CMR-based scores, which seem to have a markedly high power to predict ventricular arrhythmia (VA) and SCD in HF patients eligible for prophylactic ICD [12,13,14].

## 2. Prophylactic ICD Benefit Scores

### 2.1. MADIT-II Trial-Based Risk Stratification Score (MRSS)

One of the first scores developed was the MADIT II Risk Stratification Score (MRSS) [6], based on 1230 MADIT-II ischemic cardiomyopathy (ICM) patients who were randomized to HF medical therapy + ICD (“ICD arm”) versus medical therapy alone (“control” arm) [1], The score included five clinical parameters including age >70, blood urea nitrogen (BUN) >26 mg/dL, QRS width >120 ms, presence of atrial fibrillation (AF), and New York Heart Association (NYHA) functional classification > 2. Based on these parameters, MADIT patients were categorized into low (0 parameters), intermediate (1–5 parameters), and very high-risk (VHR) subgroups, defined by BUN > 50 mg/dL or creatinine > 2.5 mg/dL. Comparing the survival of the “control” versus “ICD” arms in each of these risk categories during the two-year follow-up (F/U) period revealed a significant ICD survival benefit only in the intermediate-risk subgroup [6], An eight-year F/U study of these same patients, categorized into low- (MRSS 0), intermediate- (MRSS 1–2), and high- (MRSS 3–5) risk subgroups showed a significant ICD survival benefit in both the low- and intermediate-risk subgroups [7]. These results were explained by arrhythmic-induced mortality in the low and intermediate subgroups, which outweighed non-arrhythmic mortality [6,7]. Notably, the low annual arrhythmic risk in the MRSS low-risk subgroup accumulated over the years and became apparent after prolonged F/U, explaining the non-significant benefit after a two-year F/U, which became significant after the eight-year F/U period. In contrast, the high and VHR subgroups (16.7% of all patients) did not show any ICD survival benefit due to the dominant non-arrhythmic death in these subgroups, where patients died from other co-morbidities before receiving any appropriate ICD therapy [6,7].

Notably, MRSS was based on a single study of American ICM patients, who were treated and followed meticulously, and thus may not represent real-world HF patients. Moreover, the MADIT study was done before the cardiac resynchronization therapy (CRT) era. Accordingly, MRSS was validated in a few studies including real-world HF patients [9,15,16]. The first of these studies included a cohort of 380 consecutive HF patients who were implanted with a primary prevention ICD in a single American center [12]. Patients were categorized according to MRSS subgroups, showing a significant incremental incidence of all-cause mortality in higher-risk subgroups (*p* < 0.001), although their incidence of appropriate ICD therapy was similar (*p* = 0.2), suggesting MRSS could not delineate arrhythmic risk in a real-world setting [15].

The second validation study included 2485 primary prevention ICD/CRTD patients from a multicenter French registry (ischemic + non-ischemic), who were categorized according to MRSS [16]. Both appropriate ICD therapies (ATP + shocks) and overall mortality were evaluated, revealing similar appropriate ICD therapy incidence (*p* = 0.9), and a significantly increased overall and non-arrhythmic mortality (including both non-arrhythmic cardiovascular death and non-cardiovascular death) among patients with higher MRSS scores (*p* < 0.001). The authors suggested that MRSS could predict ICD survival benefit in a real-world setting, including both ischemic and non-ischemic HF patients with and without CRT, by identifying patients at high risk of non-arrhythmic mortality and hence with a reduced ICD survival benefit [16].

The third validation study included a nationwide Israeli registry of 2177 HF patients with a primary prevention ICD/CRTD device, with a five-year F/U period [17]. As with prior validation studies, the incidence of appropriate ICD therapy was similar among MRSS subgroups (*p* = 0.8), with incremental overall mortality among higher-risk subgroups (*p* < 0.001). A competing-risk analysis of arrhythmic versus non-arrhythmic death was used to evaluate potential ICD survival benefit in the absence of a control group. Arrhythmic death was indirectly evaluated by appropriate ICD therapy, and non-arrhythmic death was defined as death despite ICD without prior sustained VA episodes. This competing risk analysis revealed a significantly reduced potential ICD survival benefit among higher-risk subgroups [17], recapitulating the original MRSS studies results [6,7] which showed a significant ICD survival benefit in the low and intermediate MRSS subgroups and minimal or absent benefit in the high and VHR subgroups.

Overall, MRSS was validated in more than 5000 real-world HF patients, including ICM and non-ischemic cardiomyopathy (NICM) etiologies with and without CRT. All studies revealed a similar arrhythmic incidence with a significantly increased non-arrhythmic mortality in higher MRSS subgroups. These studies suggest a minimal ICD benefit in the MRSS high-risk subgroup, given its dominant non-arrhythmic mortality exceeding the arrhythmic risk.

### 2.2. MADIT-ICD Benefit Score

The MADIT-ICD Benefit Score was developed based on 4500 primary prevention ICD patients incorporated within all MADIT studies, where ICD survival benefit was directly proven by comparison with a control group [18]. In this study two separate scoring systems were developed—one to predict fast sustained VA, and the other to predict non-arrhythmic mortality, defined by death despite ICD without prior sustained VA. Both scores included simple and easily measured clinical parameters such as age, prior MI, NYHA class, etc. The two scores were then combined to create three subgroups, including a low-benefit subgroup composed of low VA risk and high non-arrhythmic death risk, an intermediate benefit subgroup composed of low VA and low non-arrhythmic mortality risk or high VA along with high non-arrhythmic death risk, and high benefit subgroup composed of high VA risk and low non-arrhythmic death risk. Implementing this combined score on all MADIT patients showed a significant three-fold risk for VA compared to non-arrhythmic death (20% versus 7%; *p* < 0.001) in the high benefit subgroup; a lower but still increased VA risk (15% versus 9%; *p* < 0.01) in the intermediate benefit subgroup; and a similar VA and non-arrhythmic death risk (11% versus 12%; *p* = 0.4) in the low benefit subgroup [18]. The score was externally validated using the RAID Trial [19], including 1000 primary prevention ICD patients, showing good ability to predict both VA and non-arrhythmic death risk (C score 0.7 for both) [18].

### 2.3. Seattle Heart Failure Risk Model (SHFM) and Seattle Proportional Risk Model (SPRM)

The Seattle HF Model (SHFM) and the derived Seattle Proportional Risk Model (SPRM) are probably the most famous and explored models, developed to predict overall mortality and SCD risk among HF patients with reduced EF. The SHFM was developed using a cohort of 1125 HF patients with a reduced EF from the Prospective Randomized Amlodipine Survival Evaluation (PRAISE) Trial and was prospectively validated by five other cohorts, including 9900 HF patients (both with and without an ICD) [20]. Using the Cox multivariate model, various predictors for mortality were found along with their associated hazard ratios. The score comprised 24 parameters, including clinical parameters, lab results, medications, and devices (Permanent Pacemaker (PPM)/ICD/CRTD) used. The clinical parameters included age, gender, weight, NYHA class, LVEF, ischemic etiology of HF, and systolic BP. Medication-related parameters included the use of Angiotensin-converting enzyme inhibitors (ACE-I), Angiotensin receptor blockers (ARB), beta-blockers (BB), diuretics, and statins; lab parameters included serum sodium, cholesterol, WBC, % lymphocytes, hemoglobin, and uric acid. Importantly, in contrast with the clinical variables in which the hazard ratio was evaluated by the multivariable model from the PRAISE rail cohort, the hazard ratios for a subset of medications and devices were estimated from prior published literature and were not measured directly. The correlation between the model-predicted three-year survival and the actual one in the PRAISE derivation cohort was 0.99. The SHFM was then applied to the 9900 validation cohort patients, including a wide range of countries, origins, ages, NYHA symptoms, and LVEF. The correlation of the one-, two- and three-year predicted and actual survival was 0.97 by C statistic, and the overall Receiver Operating Characteristic (ROC) Area Under Curve (AUC) between SHFM predicted and the validation cohort actual survival was 0.73 [20].

Although the score uses easily obtained parameters, its calculation is not straight-forward, involving 14 continuous variables and 10 categorical ones, with the need to multiply each parameter by the natural log of its HR, making it impractical for calculation by-hand. Accordingly, a dedicated website calculator was developed for this purpose.

Thereafter, the association between SHFM predicted overall mortality and ICD survival benefit was tested. The driven hypothesis was that sicker patients with higher SHFM-predicted mortality die mostly due to non-SCD etiologies, resulting from end-stage HF leading to pump failure, as well as various non-cardiac comorbidities (such as diabetes, chronic renal failure (CRF), cerebrovascular accident (CVA), etc.). Moreover, in contrast with lethal VA which may underlie SCD in SHFM low-risk subgroups, in higher-risk subgroups the SCD might be dominated by asystole, electromechanical dissociation, or pulmonary emboli. Thus, these high-risk SHFM patients would benefit minimally from ICD. Lower-risk patients, in whom a higher ratio of SCD/non-SCD is expected and their SCD is dominated by lethal VA, would benefit more from ICD.

Accordingly, a new trial evaluated SHFM’s ability to predict SCD/non-SCD ratio among HF patients [21]. The trial was based on 2500 Sudden Cardiac Death in Heart Failure Trial (SCD-HeFT) patients [22], including symptomatic NYHA II-III HF patients with EF ≤ 35% who were randomized to control arm (HF medications only), Amiodarone, or ICD. Due to missing data regarding some of the SHFM parameters, a modification was used, named SHFM-D (SHFM differential ICD benefit score), including the following 14 parameters: Age, gender, NYHA, EF, ischemic etiology, SBP, Ace-i/ARB use, BB, Carvedilol, statin, Digoxin, Fusid dose, serum creatinine, and sodium. SCD-HeFT control and ICD arm patients were divided into five equal-size quartiles according to their SHFM-D-predicted 4-year mortality risk. In the control arm, while the 4-year mortality increased from 12% in the low-risk quartile to 50% in the highest-risk quartile, the SCD/non-SCD ratio decreased from 52% in the low-risk quartile to 24% in the highest-risk quartile. Thereafter, comparing the survival of ICD and control arm patients in each of the quartiles showed that ICD decreased SCD by 88% and total mortality by 54% in the low-risk quartile, while in the highest-risk quartile, ICD decreased SCD by 20% only and did not decrease overall mortality (*p* = 0.014) [21]. Overall, the trial suggested that apart from overall mortality in HF patients, the SHFM could also predict ICD survival benefit. Thus, SHFM lower-risk patients are suggested to have increased SCD/non-SCD ratio, and their SCD is driven by lethal VA, resulting in a significant ICD survival benefit, while SHFM higher-risk patients are suggested to have decreased SCD/non-SCD ratio and their SCD is driven mostly by non-arrhythmic mortality, resulting in a minimal if any ICD survival benefit [21]. Notably, similar to MRSS, the SHFM-D ability to predict ICD survival benefit was proven directly, based on a control group of patients without an ICD.

The above paved the way for the Seattle Proportional Risk Model (SPRM) [23], which was developed to predict the proportion of SCD/non-SCD in HF patients. Using multivariable logistic regression analysis on a cohort of 10,000 HF patients with reduced EF without an ICD, 10 parameters predicting a higher SCD/non-SCD ratio were found, including younger age, male, NYHA 1–2, lower EF, higher BMI, normal creatinine, serum sodium > 138, no diabetes mellitus, SBP~140 mmHg, and Digoxin use. Applying this model on a new cohort of 1950 symptomatic HF patients with reduced EF from the Heart Failure: A Controlled Trial Investigating Outcomes of Exercise Training (HF-ACTION) Trial [24] (half of which with an ICD), revealed a good correlation between the model predicted and actual SCD/non-SCD ratio in the placebo (no ICD) subgroup, with ROC AUC of 0.65. The association of SPRM quartiles with ICD survival benefit was then evaluated by categorizing both the placebo and ICD subgroups according to SPRM quartiles, revealing a significantly increased ICD survival benefit in higher SPRM quartiles, with 23% and 64% mortality reduction in SPRM lowest and highest quartiles, respectively (*p* = 0.001) [23].

### 2.4. Heart Failure Meta-Score

The heart failure Meta-score (Figure 1) was developed based on evidence based predictors for overall mortality among HF patients with ICD, as revealed in a meta-analysis of 72 studies including more than 250,000 HF patients with ICD (50–90% of which had primary prevention ICD while others had secondary prevention implant, and about half of studies included CRTD patients as well) [25]. Similar with SHFM, based on multivariate regression analyses, the hazard ratio of each parameter was calculated and thereafter the score itself is based on multiplication of each predictor by the natural log of its hazard ratio. The score consists of 14 predictors, of which 3 are continuous ones including age, LVEF, and glomerular filtration rate, 1 categorical predictor- NYHA class and 10 dichotomous ones, including: gender, diabetes, COPD, peripheral vascular disease, presence/absence of AF, QRS width, ischemic cardiomyopathy, prior HF hospitalization, prior VA episodes, and ICD shocks (both appropriate and inappropriate). The score was validated among the Ontario provincial database including 9860 HF patients, of which 7380 had LVEF ≤ 35%), most of which on optimal medical therapy including beta-blockers, ACE inhibitors or ARB and mineralocorticoid receptor antagonists. A third of the validation cohort had a secondary preventioin ICD and 28% had a CRTD. The model has shown remarkable prediction ability of overall mortality with observed and predicted 3-year mortality of 82% and 85%, respectively [25]. Importantly, the development process of the meta-score enables incorporation of new predictors when evidence accumulates, leading to re-adjustment and potentially, increasing predictive capabilities.

### 2.5. FADES Score

The score was developed among a single-center cohort of 900 ischemic HF patients who were implanted with a primary prevention ICD (49% with CRT) and were followed regularly every 3–6 months, documenting appropriate ICD therapies and mortality [26]. The study’s primary endpoint was death despite ICD without prior appropriate ICD therapy, representing non-arrhythmic death and absence of ICD benefit. During the study’s 2-year F/U period, 191 patients (21%) received appropriate ICD therapy, and 150 (17%) died, of which 114 (76%) died without receiving any prior appropriate ICD therapy. Based on multivariable analysis, a few predictors for the primary endpoint were identified, including advanced age > 75 years, DM, NYHA class ≥III, EF ≤ 25%, and history of smoking. Based on these predictors, the FADES (NYHA Functional class, Age, Diabetes, Ejection fraction, and Smoking) scoring was developed (Table 1). Few risk subgroups were delineated, including low (0–1.5 points), intermediate (2–2.5 points), and high-risk (3–5.5 points) subgroups, yielding a 5-year cumulative risk for the primary endpoint of 10%, 17%, and 41% in the low-, intermediate-, and high-risk subgroups, respectively (*p* < 0.01). A good correlation was found between the risk-based prediction and actual death without appropriate ICD therapy (AUC 0.73) [26]. Notably, there was no significant difference between the risk subgroups regarding the incidence of appropriate ICD therapy.

### 2.6. SHOCKED Score

This score was developed to predict overall mortality among real-world clinical practice patients with prophylactic ICD [27]. The score was developed based on 17,990 Medicare patients with prophylactic ICD (including NYHA II-III ischemic and non-ischemic patients with EF ≤35%, ischemic NYHA I patients with EF≤ 30%, and post-MI patients with EF ≤ 40% with NSVT and induced sustained VT/VF in EPS). Using the Cox proportional hazards regression model, the strongest predictors for overall mortality were identified, including age > 75, NYHA III, EF≤ 20%, AF, chronic pulmonary disease, CRF, and DM. Each predictor was assigned points, reflecting its HR for overall mortality, and the risk score was calculated by summation of the points attributed to each of the seven predictors (Table 1). The score was validated among a separate Medicare cohort of 27,890 patients with prophylactic ICD. The relation between the risk predicted and actual mortality, evaluated via C-statistics, was 0.75 and 0.74 for the development and validation cohorts, respectively. When divided into five risk quartiles, the 3-year mortality rate increased from 11% to 58% in the lowest and highest risk quartiles, respectively. Notably, CRF was the strongest predictor for overall mortality among these patients [27].

### 2.7. PACE and Charlson Comorbidity Index-Based Scores

The PACE Score [28] was developed to predict early (<1 year) mortality despite ICD, trying to accommodate current guidelines, which recommend avoiding ICD implantation in patients with a life expectancy of <1 year [3,4]. Using a cohort of 2717 ICD/CRTD recipients (75% primary prevention and 25% secondary prevention) from 3 large tertiary hospitals, one-third of the cohort was randomly selected to consist of the prediction cohort, from which the score was developed, and the other two-thirds served as a validation cohort. Using stepwise logistic regression on the prediction cohort, four “PACE” predictors were identified, including peripheral arterial disease, age ≥ 70, creatinine ≥ 2 mg/dL, and EF ≤20% (Table 1). The PACE Score accurately predicted 1-year mortality among the validation cohort (c-statistic 0.79). Patients with a PACE score of 0, 1, 2, 3, and 4–5 had a one-year mortality of 1.7%, 4%, 6.9%, 15.5%, and 18.2%, respectively (*p* < 0.001). A marked dichotomy of 1-year mortality was found between patients with PACE score ≥ 3 (6% of the cohort) versus those with PACE < 3, with a 1-year mortality of 16.5% versus 3.5%, respectively (*p* < 0.001) [28]. Similar to the SHOCKED score, chronic renal failure was found to be the strongest predictor for early mortality. Last but not least, the Charlson comorbidity index (CCI) is one of the famous scores to assess patient frailty [29]. The score is a simple summation of patient co-morbidities, including age > 80, AIDS, CVA, COPD, CHF, connective tissue disease, DM, dementia, liver disease, peptic disease, malignancy, peripheral vascular disease, and CRF. Using the CCI to predict early mortality among ICD recipients showed a 1-year mortality of 5% versus 78% in the low (CCI score 0) and high (CCI score ≥5) scores, respectively. Moreover, patients with high CCI also had a significantly reduced incidence of appropriate ICD therapy, suggesting these patients have a high risk for non-arrhythmic death and low if any ICD survival benefit [29].

## 3. Comparison between Models

Few studies have tried to compare between the above scores (Figure 2) [9,25,30,31]. The first [9], compared between FADES, MRSS, and SHFM-D models’ performance in predicting mortality despite ICD without prior appropriate ICD therapy (ICD non-benefit), and their ability to discriminate between ICD non-benefit and appropriate ICD therapy (ICD benefit), among a cohort of 1970 HF patients who were implanted prophylactic ICD (58% with CRT). All three models were predictive of ICD non-benefit (*p* < 0.001 in all three) and their predictive performance, evaluated via C-statistics, was 0.66, 0.69, and 0.75 for the FADES, MRSS, and SHFM-D, respectively. Regarding their discrimination performance, highest-risk category patients in both SHFM and MRSS models had 1.7 times higher risk for ICD non-benefit than ICD benefit, while highest-risk category patients in FADES were as likely to experience ICD non-benefit as ICD benefit. The study suggests that SHFM-D is superior to MRSS and FADES due to its remarkable ability to predict ICD non-benefit and discriminate between ICD survival benefit and ICD non-benefit [9].

The second study [25], compared the discrimination of the HF meta-score [25] with the SHFM [20] and SHOCKED [27] scores for overall mortality in HF patients with ICD (measured by C-statistics) was 0.72, 0.73, and 0.7 for the scores, respectively. Notably the discrimination of meta-score was not significant from that of SHFM and both the meta-score and SHFM were significantly better than the SHOCKED score [27].

The Third study compared SHFM, MRSS, and CCI risk scores in predicting 5-year mortality in a cohort of 823 patients implanted with a prophylactic ICD/CRTD [30]. The actual 5-year mortality among the cohort was 21%,. The performance of the three models in predicting actual 5-year mortality for ICD/CRTD patients, evaluated via C statistics, was 0.71/0.73, 0.61/0.7, and 0.65/0.66 for the SHFM, MRSS, and CCI models, respectively. Overall, the SHFM had the best mortality-predicting performance among both ICD and CRTD recipients. Evaluating the impact of the various predictors of all models on the overall mortality of the entire cohort, via multivariable analysis, revealed that age > 70, NYHA > 2 class, chronic renal failure, and cancer were the strongest predictors [30]. Notably, as in many prior studies [6,23,24], chronic renal failure was the strongest predictor for overall mortality.

The Fourth study compared MRSS, FADES, PACE, and SHOCKED, in predicting 4-year mortality among 916 prophylactic ICD patients (ischemic and non-ischemic HF) from 15 Spanish hospitals [31]. Categorizing patients according to all four risk scores showed a significantly increased 4-year mortality in high-risk categories in all four scores (*p* < 0.001). The correlation between the predicted and actual 4-year mortality, measured via C statistic, was 0.66, 0.63, 0.61, and 0.64 for the MRSS, FADES, PACE, and SHOCKED scores, respectively [31].

Overall, although none of the risk scores could replace current guidelines for prophylactic ICD implantation, they all raise major points to consider, including (1) A subset of prophylactic ICD-eligible patients have an extremely high chance of early mortality after ICD implantation. For example, 20% of prophylactic ICD eligible patients in SHFM-D high-risk category had 1-year mortality~30% [18], and some patients in CCI study had 1-year mortality of 78% [28]. (2) Multiple risk score studies have consistently shown a high ICD benefit/ICD non benefit ratio, reflected by the ratio of appropriate ICD therapy compared with that of death without prior appropriate therapy, in low-risk categories. In contrast, a low ICD benefit/ ICD non-benefit ratio was found in high-risk category patients, suggesting these patients have the least ICD survival benefit [9,15,17,21,25]. (3) Both MRSS and SHFM-D models, based on randomized trials with a control group and validated on large ‘real-world’ cohorts, have proven no ICD benefit in non-negligible subgroups. Thus, no survival benefit was shown in 20% of patients consisting of the SHFM-D high-risk category [21], and in 16.7% of patients consisting of the MRSS high-risk category [7]. A suggested use of the various scores to evaluate early mortality and mortality without prior appropriate ICD interventions (namely ‘ICD non-benefit’) is shown in the graphical abstract. 

Due to the limitation of current scores future research is needed to re-define CHF subgroups which would benefit from prophylactic ICD in contemporary era including new medications and using novel risk factors for SCD, other than the “LVEF stand alone” one. One of these researches for ICM is the PROFID project [32] which was initiated in Dec 2023 and intended to include 2 phases: The first- “PROFID reduced” trial randomizing post MI patients with EF ≤ 35% to contemporary optimal medical therapy (OMT) versus OMT+prophylactic ICD with a primary endpoint of all-cause death within 2.5 years. The second phase-:”PROFID preserved” to evaluate new SCD risk factors, including post MI patients with LVEF > 35% with another risk factor who will be randomized between OMT versus OMT+prophylactic ICD [32]. As for NICM patients, the BRITISH randomized trial [33] is planned to include 1252 NICM patients with LVEF ≤ 35% along with a CMR scar, to evaluate CMR ability to better delineate patients with increased SCD risk. The trial will randomize such patients between

## 4. Prophylactic ICD in Pediatrics

Prophylactic ICD has an important role in the pediatric population, in a range of inherited arrhythmia syndromes (‘channelopathies’) as Long QT and Brugarda syndromes, as well as among cardiomyopathies including hypertrophic CM (‘HCM’) which results from sarcomeric protein mutations, arrhythmogenic right ventricular CM (‘ARVD’) caused by desmosomal proteins, and dilated CM patients, some of which are attributed to genetic mutations, usually in cardiac cytoskeletal proteins [34]. Prophylactic ICD use in various channelopathies is out of the scope of the current review. As for NICM or dilated CM, the risk stratification for SCD is still evolving [34]. In hypertrophic CM (HCM), SCD risk stratification is less studied compared with adults. However, unexplained syncope, severe LVH, prior non-sustained VT (NSVT), and family history of SCD are considered by many as major risk factors that can justify prophylactic ICD [34]. Notably, a pediatric-oriented “HCM risk-kids” was developed, including LV wall thickness, LA diameter, unexplained syncope, prior NSVT, and maximal LVOT gradient. The use of this score yielded an AUC of 0.75 for predicting a major arrhythmic event in pediatric patients with HCM [34].

## 5. CMR-Based Scores Predicting VA and SCD

Multiple recent publications show CMR can reliably detect myocardial fibrosis or scar tissue, known to serve as substrate for initiation and maintenance of VA and may predict future VA events among HF patients. A few of the pivot trials in this field are presented hereby [12,13,14].

One of the initial pivot studies used CMR-late gadolinium enhancement (LGE) to detect myocardial scars among 1165 consecutive non-ischemic HF patients from two tertiary high-volume CMR centers. In this study, LGE was found as an independent and robust predictor for sustained VA or SCD during a median 3-year F/U period, regardless of patients’ EF [12]. A simple algorithm combing LGE results (considering LGE location, distribution, and extent) and EF (divided to ≤20%, 21–35%, >35%) was significantly superior to the EF 35% “stand-alone” cutoff risk stratification method (ROC AUC of 0.82 versus 0.7; *p* < 0.001), which is the method used world-wide and endorsed by current guidelines to decide on primary prevention ICD [28]. Importantly, using this combined LGE-EF algorithm, patients with EF ≤ 35% with negative LGE were found to be at low risk for future VA or SCD, while patients with EF > 35% with positive high-risk LGE distribution were found to be at high-risk for future VA or SCD [12]. Although ICD benefit was not evaluated directly in this study, it was suggested to imply such benefit among patients found to be at high risk for future VA events.

Moving a step further, the use of CMR-LGE was evaluated among 700 HF patients (408 ischemic, 292 non-ischemic), in whom the CMR was performed just before ICD or CRTD implantation [13]. The study showed that all cases with eventual SCD had myocardial fibrosis on CMR, and there was no case of SCD in patients without myocardial fibrosis. Moreover, only 2.4% of cases who eventually had a composite arrhythmic endpoint including SCD, resuscitated SCD, sustained VA, and appropriate ICD therapy, had no myocardial fibrosis on CMR. Accordingly, myocardial fibrosis assessment via CMR had a negative predictive value of 100% for SCD and 98.6% for the composite arrhythmic endpoint. On multivariable analysis for both SCD and composite endpoint, including age, HF etiology, prior myocardial infarction, DM, medications used, QRS duration, EF, and the presence or absence of myocardial fibrosis on CMR, myocardial fibrosis was the only parameter found to be independently associated with both endpoints. Moreover, among patients with myocardial fibrosis, a larger extent of the “gray-zone” area, presenting a mixture of viable and non-viable myocardium, was found to be significantly associated with increased risk for composite arrhythmic endpoint. Using CMR among ICD recipients to assess presence of myocardial fibrosis and extent of “gray-zone” area, one could delineate subgroups with a significantly different 7-year risk for composite arrhythmic endpoint, including low-risk subgroup (0.14% annual risk), defined by absence of myocardial fibrosis, an intermediate-risk subgroup (1.2% annual risk) in the presence of myocardial fibrosis with moderate “gray-zone” extent (<17 g), and a high-risk subgroup (4.5% annual risk) among patients with myocardial fibrosis and large “gray-zone” area (>17 g). Overall, in this study, CMR was shown to be a strong predictor for appropriate ICD use among HF patients, in contrast with EF. Moreover, CMR could reliably identify a non-negligible subgroup (30% of ICD recipients in this study) with a very low risk for future arrhythmic events, in whom an ICD benefit would be questionable [13].

A more modern approach considers not only the presence and extent of a scar but also the scar’s “architecture”. Such an approach was recently used in a study [14] evaluating 200 HF patients (ICM+NICM), who underwent CMR-LGE before primary prevention ICD implant, using dedicated ADAS-3D software (ADAS LV Medicine version 2.14, Barcelona, Spain) which could automatically identify scar, border zone, core, and ‘conducting channels’. Scar mass, border zone area, core mass, and ‘conducting channels’ were all significantly associated with eventual ICD appropriate therapy. Importantly, the presence of “conducting channels”, a novel feature of the scar architecture, was independently associated with appropriate ICD therapies (HR 4.17). The authors concluded that scar characteristics analyzed by LGE-CMR are strong predictors of ICD appropriate therapy, and the absence of channels with a scar mass <10 g was associated with a very low risk of future VA [14].

Over the past decade, multiparametric mapping analysis has become an integral component of the CMR protocol. This technique serves not only as a crucial diagnostic tool but also as a valuable prognostic instrument. Several studies have underscored the significance of multiparametric mapping in CMR for predicting VA by detecting pro-arrhythmic substrates, including myocardial scar and interstitial fibrosis, which could both serve as a substrate for re-entry arrhythmias [35,36,37,38]. Chen et al [35]. demonstrated that non-contrast T1 mapping CMR can effectively characterize myocardial scar tissue, providing essential prognostic information for predicting VA in patients with both ICM and NICM who have undergone ICD implantation after the CMR study. Further advancing the field, Nakamori et al [36]. showed that native T1 mapping can assess tissue heterogeneity in patients with dilated cardiomyopathy (DCM),with an increased arrhythmic risk even in the absence of LGE, suggesting that T1 mapping could add an arrhythmic predictive value on top of LGE. Additionally, Claridge’s work [37] supported these findings in NICM patients but did not observe the same effect in ICM patients. A recent comprehensive review [38] has explored the current evidence linking mapping results with the risk of VA, highlighting the future potential of this technique in various cardiomyopathies.

## 6. Genetic Mutation for Arrhythmia Risk Prediction

As of today, numerous studies have pointed to various gene mutations which contribute to development and severity of non-ischemic cardiomyopathy (NICM) phenotype and some which contribute specifically to an increased arrhythmia risk. A recent editorial dedicated to predicting arrhythmic risk via gene mutations in NICM [39] revealed few genes which seem to predict a significantly increased risk for ventricular arrhythmia and SCD. Among these are the well known Lamin A/C and Filamin C mutations, as well as a group of desmosomal mutations which contribute to a dominant LV cardiomyopathy (without classic ARVD manifestations) with an exceptionally high arrhythmic risk [39]. Importantly, a recent study of Gigli et al. [40] found relevant genetic mutations in 37% of a cohort including 482 patients with NICM, suggesting that gene mutations could be found in a significant percent of these patients. Noteworthy, a thorough evaluation of NICM patients including detailed review, physical examination, imaging including CMR, and genetic testing was recently shown to enable personalized arrhythmic risk stratification of maximal accuracy [41].

## 7. Summary

To date, multiple risk scores which have been validated on thousands of ischemic and non-ischemic real-world HF patients with prophylactic ICD, evaluating major endpoints relevant to these specific patients, including early < 1-year mortality after ICD implant, death despite ICD without any prior appropriate ICD intervention, and cumulative incidence of appropriate ICD therapy (Table 1). According to these scores, a significant proportion of patients who are eligible for primary prevention devices according to current guidelines, would have a minimal ICD survival benefit, due to extremely high non-arrhythmic mortality (due to multiple other co-morbidities and end-stage HF), a very low incidence of VA events, and a low rate of SCD dominated by non-shockable rhythm (ex. Asystole). Few of the scores, developed from randomized trials comparing control group (HF medications only) to ICD group (HF medications+ICD), prove no actual benefit in subgroups with a high risk of non-arrhythmic mortality, as elderly patrients with advanced renal disease any advanced CHF with NYHA > 2. Although these scores have a modest-moderate predictability for VA (with AUC < 0.75 for most) and thus could not be implemented in current guidelines, these scores could still help physicians in their recommendation for prophylactic ICD implants in borderline cases such as elderly patients and those with multiple advanced co-morbidities, which were excluded from many pivot trials of prophylactic ICD (ex. patients with cerebro-vascular disease or advanced non-cardiac co-morbidities who were likely to die within the trial period, were excluded from MADIT II trail [1]). Other scores could reliably predict an early (<1-year) mortality despite ICD among a non-negligible proportion of prophylactic ICD candidates, in whom an ICD should not be implanted even according to current guidelines [3,4]. This early mortality is particularly relevant to “real-world” patients who are usually older and with multiple co-morbidities, compared with typical study patients. CMR was recently shown to reliably detect myocardial scar and specific scar architecture, as border zone and ‘channels’. Thus, delineating prophylactic ICD patients with a low- versus high-risk for SCD and VA episodes, overweighing “stand-alone EF”, upon which current guidelines are based. Last but not least, future research is needed to re-define CHF subgroups which would benefit from prophylactic ICD in contemporary era including new medications and using novel risk factors for SCD, other than the “LVEF stand alone” one. 

## Figures and Tables

**Figure 1 jcm-13-05307-f001:**
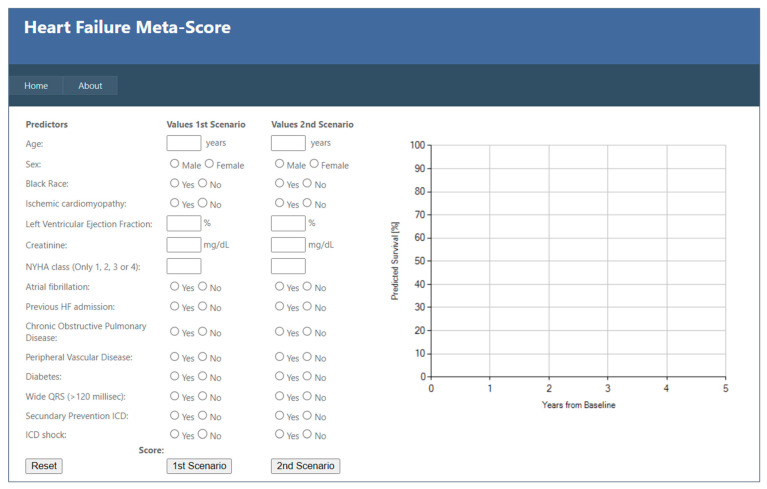
The Heart Failure Meta-score built in a user-friendly online format to estimate survival during a 10-year follow-up in 2 different scenarios or patients simultaneously. This Meta-Score table is available at http://www.hfmetascore.org/HeartScore.aspx. accessed on 24 August 2024.

**Figure 2 jcm-13-05307-f002:**
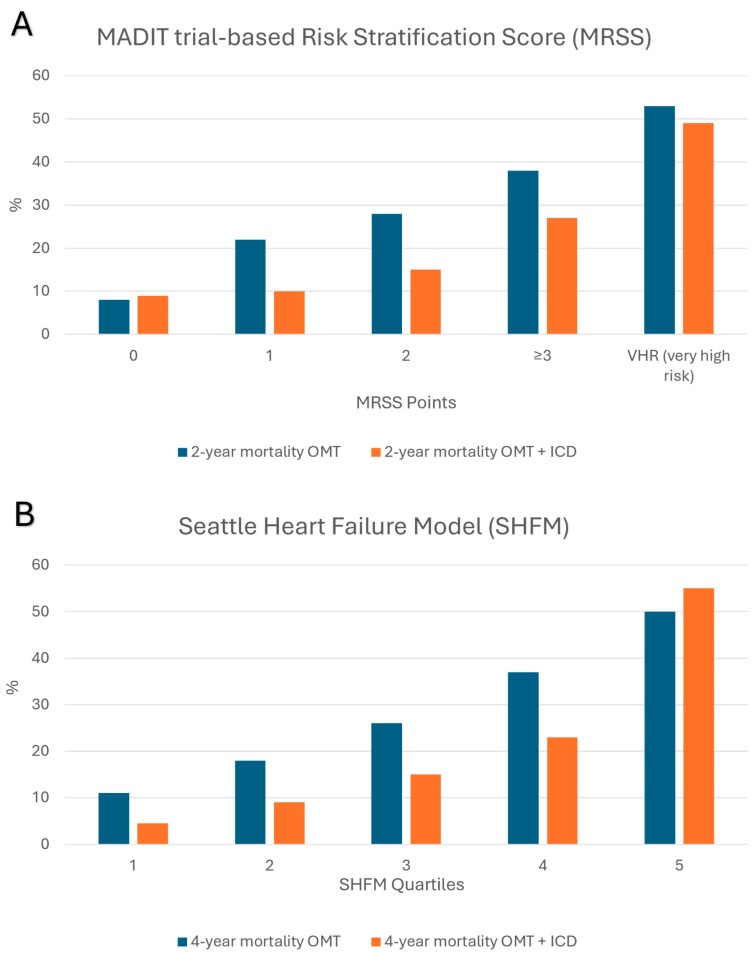
Comparison of overall mortality with optimal medical therapy (OMT) versus OMT + prophylactic ICD according to MRSS risk subgroups (A) and SHFM quartiles (B), revealing an ICD survival benefit among MRSS and SHFM intermediate-risk subgroups and absence of ICD survival benefit in high-risk subgroups (MRSS VHR subgroup and SHFM quartile #5). The figures are based on published data of MRSS [6] and SHFM [21].

**Table 1 jcm-13-05307-t001:** Main scores to predict prophylactic ICD survival benefit.

Parameters/Score	MRSS [6,7]	SHFM-D [18]	FADES [23]	SHOCKED [24]	PACE [25]
Parameters included	Points per predictor Age > 70 1 BUN ** >26 mg/dL 1 QRS > 120 ms 1 Atrial fibrillation 1 NYHA class > 2 1	Complex algorithm * Age, Gender, SBP, ischemic HF etiology, NYHA class, LVEF, BB, ACEi/ARB, Digoxin, Statin, Fusid dose, Serum Creatinine, Serum Sodium	Points per predictor Age 65–74 0.5 Age ≥ 75 2 NYHA ≥ 3 1 LVEF ≤ 25% 1 DM 1 Smoking 1	Points per predictor Age > 75 62 NYHA III 36 LVEF ≤ 20% 28 Atrial fibrillation 27 CRF *** 100 COPD *** 62 Diabetes Mellitus 41	Points per predictor PAD *** 1 Age ≥ 70 1 Crea t *** ≥ 2 mg/dL 2 LVEF ≤ 20% 1
Risk ca > 26egories	Low 0 Intermediate 1–2 High 3–5 VHR-BUN > 50 mg/dL /Creatinine > 2.5 mg/dL	Quintiles *	Low 0–1.5 Intermediate 2–2.5 High 3–6	Quintiles	Per points (0–5)
Risk endpoint	8-year ICD survival benefit compared with control w/o ICD	4-year ICD survival benefit compared with control w/o ICD	5-year death w/o appropriate ICD therapy	1 to 4-year overall mortality	1-year overall mortality
Endpoint incidence by risk category (subgroups with least predicted ICD survival benefit marked in red)	8-year relative mortality reduction Low-risk 48%; *p* < 0.001 Intermed 34%; *p* < 0.001 High-risk 16%, *p* = 0.25	4-year relative mortality reduction (quintiles) 54% 43% 37% 30% 0%	5-year death w/o appropriate ICD therapy Low-risk 10% Intermediate-risk 17% High-risk 41%	3-year mortality (quintiles) 11% 21% 28% 40% 58%	year overall mortality(score points) 1.7% 4% 6.9% 15.5% 4–5 18.2%
Significant difference of ICD appropriate therapy between categories	No	No	No	Not Evaluated	Not Evaluated
Based on control group-HF patients w/o ICD	Yes (MADIT II)	Yes (SCD-HeFT)	No	No	No
Based on which HF etiology	Developed on Ischemic HF but validated upon ischemic + non-ischemic	Developed on ischemic + non-ischemic	Ischemic only	Developed on ischemic+non-ischemic	Developed on ischemic + non-ischemic
Based on prophylactic ICD/CRTD	Developed on ICD only but validated among both ICD/CRTD	Developed on ICD only but validated among both ICD/CRTD	Developed on ICD + CRTD	Developed and validated on ICD only	Developed and validated among both ICD/CRTD
External validation	Multicenter French registry (n = 2485; with prophylactic ICD/CRT) [13] Israeli nationwide registry (n = 2177;with prophylactic ICD/CRT) [14]	5 different studies including > 10,000 HF pt with and w/o prophylactic ICD [17]	Single center cohort (n = 1970; with prophylactic ICD) [9]	Medicare cohort (n = 27,890; with prophylactic ICD) [24]	3-hospital cohort (n = 1812; with prophylactic ICD) [25]

* in contrast with FADES and MRSS which have a simple straightforward scoring, the SHFM-D is calculated by a complex algorithm, where each predictor is multiplied by the natural log of its Hazard Ratio and then summed. ** BUN Blood Urea Nitrogen; VHR = Very High Risk. *** CRF = Chronic Renal Failure; COPD = Chronic Obstructive Pulmonary Disease; PAD = Peripheral Arterial Disease; Creat = Creatinine.

## Data Availability

Not applicable.

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
