# Peer review of "Prophylactic ICD Survival Benefit Prediction: Review and Comparison between Main Scores"

_jcm, 2024, doi:10.3390/jcm13175307_

Round 1

Reviewer 1 Report

Comments and Suggestions for Authors

The authors wrote an interesting review about clinical risk scores for predicting the benefit of primary prevention ICD therapy in patients with heart failure. They adequately identified the most important scores and compared them. References are adequate. However, I think that the translation in clinical practice in the text may be optimized: Most scores only show modest predictability (AUC of < 0.7) and therefore are not good enough to be used in clinical practice. Furthermore, clinical guidelines currently do not endorse any scores for the indication of ICD implantation. I recommend to include a pathway how to use the scores in clinical practice. For example, do the authors use the scores? Should the patient’s prefrence also be included into the decision?

Furthermore, I recommend to include an outlook to future research with emphasis on randomized studies. I think that PROFID is the most important one, but it is certainly not the only study.

 Author Response

Reviewer #1:

The authors wrote an interesting review about clinical risk scores for predicting the benefit of primary prevention ICD therapy in patients with heart failure. They adequately identified the most important scores and compared them. References are adequate. However, I think that the translation in clinical practice in the text may be optimized: Most scores only show modest predictability (AUC of < 0.7) and therefore are not good enough to be used in clinical practice. Furthermore, clinical guidelines currently do not endorse any scores for the indication of ICD implantation. I recommend to include a pathway how to use the scores in clinical practice. For example, do the authors use the scores? Should the patient’s prefrence also be included into the decision?

Answer: We thank the reviewer for this very important comment. Indeed, most of the scores show modest-moderate predictability and thus the scores could not replace current guidelines. Nevertheless, they can aid in borderline cases as elderly and CHF patients with multiple co-morbidities characterized by a dominant non-arrhythmic mortality, which were excluded from many pivot trials upon which the guidelines were based. For example, in MADIT II trial, patients with cerebro-vascular disease were excluded as well as patients with advanced non-cardiac co-morbidities which seemed to have a high likelihood of death during the trial period. We have actually suggested such use of the scores in the original manuscript (at the summary paragraph) but totally agree with the reviewer that this should be stressed and made clear throughout the manuscript. Accordingly, we revised both the abstract conclusion paragraph (page 2) and the  summary paragraph (page 15, bottom) saying that: “Although these scores have a modest-moderate predictability for VA (with AUC < 0.75 for most) and thus could not be implemented in current guidelines, these scores could still help physicians in their recommendation for prophylactic ICD implants in borderline cases such as elderly patients and those with multiple advanced co-morbidities, which were excluded from many prophylactic ICD pivot trials (ex. patients with cerebrovascular disease or advanced com-morbidities who were likely to die within the trial period, were excluded from MADIT II trail [1]).” Moreover, in line with current guidelines, we recommend using scores to assess high likelihood of early mortality (<1-year) in whom ICD should not be implanted according to both American and European guidelines. This is clearly written at the summary paragraph (page 15, bottom). Last and not least, a new graphical abstract, dedicated to a suggested clinical use of the various scores, was added to further clarify our suggestion for the use of the various scores in daily practice.

Furthermore, I recommend including an outlook to future research with emphasis on randomized studies. I think that PROFID is the most important one, but it is certainly not the only study.

Answer: In accordance with the reviewer recommendation we include in the revised extended summary paragraph (page 16) two of the leading future studies intended to increase prophylactic ICD benefit in ischemic patients (the PROFID), using contemporary CHF medications and other novel SCD risk factors, and among non-ischemic patients (BRITISH trial), by evaluating CMR scar as a novel risk factor for SCD in NICM.

Reviewer 2 Report

Comments and Suggestions for Authors

Congratulations to the authors for skillfully addressing a highly debated topic, such as the implantation of ICDs for primary prevention. The review is well-constructed but requires some revisions:

Although the manuscript references recent studies utilizing cardiac MRI (CMR) to predict ventricular arrhythmia (VA), it could be enhanced by incorporating the most recent advancements in CMR techniques and their relevance to clinical practice. For example, exploring the roles of T1 and T2 mapping or extracellular volume (ECV) measurements could offer a more complete perspective on the potential of CMR in risk stratification;

You might want to include a discussion on how genetic markers can be integrated into predicting ICD benefits. Recent studies have investigated the impact of specific genetic mutations on cardiac arrhythmias, which could be a valuable addition to the conversation on risk stratification;

Incorporating a section on the cost-effectiveness of ICD implantation relative to different risk scores could improve the manuscript. This topic is particularly relevant in light of the financial burden discussed and the varying degrees of benefit observed across different patient subgroups;

The authors should include a brief section on ICD implantation in the pediatric population, incorporating and discussing “Inherited Arrhythmias in the Pediatric Population: An Updated Overview. Medicina (Kaunas). 2024 Jan 3;60(1):94. doi: 10.3390/medicina60010094.” , given the significant prevalence of potentially fatal arrhythmias in these patients and the impact that such devices may have on this specific group;

The conclusion could be strengthened by summarizing the main insights from the various risk scores and offering more definitive recommendations for clinical practice. Adding a discussion on future research directions and the potential for incorporating AI into risk stratification would also enhance the value of the manuscript;

Authors  should add some figures to make the manuscript more captivating

Author Response

Reviewer #2:

Congratulations to the authors for skilfully addressing a highly debated topic, such as the implantation of ICDs for primary prevention. The review is well-constructed but requires some revisions:

Although the manuscript references recent studies utilizing cardiac MRI (CMR) to predict ventricular arrhythmia (VA), it could be enhanced by incorporating the most recent advancements in CMR techniques and their relevance to clinical practice. For example, exploring the roles of T1 and T2 mapping or extracellular volume (ECV) measurements could offer a more complete perspective on the potential of CMR in risk stratification;

Answer: We thank the reviewer for his positive feedback on pour review in general. In accordance with the reviewer recommendation we extended the CMR section by addressing the possible contribution of CMR multiparametric mapping for arrhythmic risk stratification in a new paragraph (page 14, 3rd paragraph).

You might want to include a discussion on how genetic markers can be integrated into predicting ICD benefits. Recent studies have investigated the impact of specific genetic mutations on cardiac arrhythmias, which could be a valuable addition to the conversation on risk stratification;

Answer: We thank the reviewer for this important remark. Accordingly, we added a new paragraph (page 15) regarding the use of genetic markers which were found to be associated with and predict ventricular arrhythmias and SCD, mostly for dilated cardiomyopathy patients.

Incorporating a section on the cost-effectiveness of ICD implantation relative to different risk scores could improve the manuscript. This topic is particularly relevant in light of the financial burden discussed and the varying degrees of benefit observed across different patient subgroups;

Answer: We agree that this is an important topic. However, given the fact that the scores are not ready for clinical implementation due to their modest predictability (see Reviewer #1 comments and our answer to the next remark regarding study conclusion), they will not change ICD cost-effectiveness and thus, this issue is not addressed in current review.

The authors should include a brief section on ICD implantation in the pediatric population, incorporating and discussing “Inherited Arrhythmias in the Pediatric Population: An Updated Overview. Medicina (Kaunas). 2024 Jan 3;60(1):94. doi: 10.3390/medicina60010094.”, given the significant prevalence of potentially fatal arrhythmias in these patients and the impact that such devices may have on this specific group;

Answer: We thank the reviewer for this important comment. Accordingly, we added a new paragraph dedicated for prophylactic ICD in pediatric patients (page 12).

The conclusion could be strengthened by summarizing the main insights from the various risk scores and offering more definitive recommendations for clinical practice. Adding a discussion on future research directions and the potential for incorporating AI into risk stratification would also enhance the value of the manuscript;

Answer: In accordance with the reviewer remark (and similar with Reviewer #1 recommendation) we extended the summary paragraph (page 15, bottom) by a clear recommendation regarding the use of the various scores in borderline cases, aiding ‘personalized’ decision making, emphasizing that they are still not ready to become implemented in guidelines. This point is further clarified in a new graphical abstract, dedicated to our suggestion of the clinical use of the various scores in daily practice. Moreover, we emphasized the need for future studies with a short note of the main studies which were recently initiated for the aim of improving prophylactic ICD candidate selection (page 16).

Authors should add some figures to make the manuscript more captivating

Answer: According to the reviewer recommendation, few figures were added to the revised manuscript. First, a graphical abstract was added, describing a suggested clinical use of the various scores for a decision-making regarding prophylactic ICD implant. Figure 1 shows an example of the META score calculator, including the various parameters comprising this score. Figure 2 shows nicely a comparison of overall mortality with optimal medical therapy (OMT) versus OMT + prophylactic ICD according to MRSS risk subgroups (A) and SHFM quartiles (B), revealing an ICD survival benefit among MRSS and SHFM intermediate-risk subgroups and absence of ICD survival benefit in high-risk subgroups (MRSS VHR subgroup and SHFM quartile #5). Thus, emphasizing a non-negligible subgroup in which prophylactic ICD does not benefit overall survival. 

Reviewer 3 Report

Comments and Suggestions for Authors

We would like to thank the authors for submitting their manuscript to our journal. The review addresses a long-debated topic regarding the predictors of Sudden Cardiac Death (SCD) in patients with Heart Failure with Reduced Ejection Fraction (HFrEF) and the subsequent indication for implantable Cardioverter Defibrillator (ICD). The following minor revisions are requested:

1) For reviews, in accordance with the journal guidelines, there is no need to divide the abstract into subsections. Please remove them.

2) The introduction should be expanded to discuss more extensively the European and American guidelines on the topic, highlighting similarities and differences.

3) The type of review being conducted should be clearly stated in the introduction (likely a narrative review), with explicit criteria for the selection of included bibliographic sources.

4) We suggest the authors create a graphical abstract summarizing the message their work intends to convey.

5) In the section dedicated to Cardiac Magnetic Resonance (CMR), reference should be made to the incremental role of mapping as a predictor of events. In this regard, we recommend the following references:

- Nakamori, Shiro et al. "T1 Mapping Tissue Heterogeneity Provides Improved Risk Stratification for ICDs Without Needing Gadolinium in Patients With Dilated Cardiomyopathy." JACC. Cardiovascular imaging vol. 13,9 (2020): 1917-1930. doi:10.1016/j.jcmg.2020.03.014;

- Lo Monaco, Maria et al. "Multiparametric Mapping via Cardiovascular Magnetic Resonance in the Risk Stratification of Ventricular Arrhythmias and Sudden Cardiac Death." Medicina (Kaunas, Lithuania) vol. 60,5 691. 24 Apr. 2024, doi:10.3390/medicina60050691

- Trimarchi, Giancarlo et al. "Transient Left Ventricular Dysfunction from Cardiomyopathies to Myocardial Viability: When and Why Cardiac Function Recovers." Biomedicines vol. 12,5 1051. 9 May. 2024, doi:10.3390/biomedicines12051051. 

These revisions will enhance the clarity and impact of the manuscript. Thank you for your attention to these suggestions.

Author Response

Reviewer #3:

We would like to thank the authors for submitting their manuscript to our journal. The review addresses a long-debated topic regarding the predictors of Sudden Cardiac Death (SCD) in patients with Heart Failure with Reduced Ejection Fraction (HFrEF) and the subsequent indication for implantable Cardioverter Defibrillator (ICD). The following minor revisions are requested:

1) For reviews, in accordance with the journal guidelines, there is no need to divide the abstract into subsections. Please remove them.

Answer: We removed subsections in the abstract.

2) The introduction should be expanded to discuss more extensively the European and American guidelines on the topic, highlighting similarities and differences.

Answer: In accordance with the reviewer comment, the revised Introduction was extended to elaborate on the ESC and ACC/AHA guidelines for both ischemic and non-ischemic CM patients (page 3).

3) The type of review being conducted should be clearly stated in the introduction (likely a narrative review), with explicit criteria for the selection of included bibliographic sources.

Answer: We added in the revised Introduction that this is a narrative review (page 3, bottom). Regarding criteria for selection of sources, we did NOT search systemically for all bibliographic sources but rather selected the main and most commonly mentioned scores, with emphasis on scores which were developed based on pivot randomized studies of HF patients treated by medications only versus medications +ICD, and have been validated in real-world cohorts. This is clearly written at the end of our Introduction. To be clear on this issue we added a sentence saying that: “Notably, no systemic bibliographic search was used for this review.” (page 4, top).

4) We suggest the authors create a graphical abstract summarizing the message their work intends to convey.

Answer: In accordance with the reviewer’s comment, we added a graphical abstract, dedicated to our suggestion for the clinical use of the various scores in daily clinical practice.

5) In the section dedicated to Cardiac Magnetic Resonance (CMR), reference should be made to the incremental role of mapping as a predictor of events. In this regard, we recommend the following references:

- Nakamori, Shiro et al. "T1 Mapping Tissue Heterogeneity Provides Improved Risk Stratification for ICDs Without Needing Gadolinium in Patients With Dilated Cardiomyopathy." JACC. Cardiovascular imaging vol. 13,9 (2020): 1917-1930. doi:10.1016/j.jcmg.2020.03.014;

- Lo Monaco, Maria et al. "Multiparametric Mapping via Cardiovascular Magnetic Resonance in the Risk Stratification of Ventricular Arrhythmias and Sudden Cardiac Death." Medicina (Kaunas, Lithuania) vol. 60,5 691. 24 Apr. 2024, doi:10.3390/medicina60050691

- Trimarchi, Giancarlo et al. "Transient Left Ventricular Dysfunction from Cardiomyopathies to Myocardial Viability: When and Why Cardiac Function Recovers." Biomedicines vol. 12,5 1051. 9 May. 2024, doi:10.3390/biomedicines12051051. 

Answer: We thank the reviewer for this important point. In accordance with the reviewer comment, a new paragraph was added (page 14, 3rd paragraph), dedicated to the added prognostic value of CMR mapping, using some of the suggested references and others as well.

Round 2

Reviewer 2 Report

Comments and Suggestions for Authors

Congratulations to the authors for having answered to all of my comments.